

# A simple method of equine limb force vector analysis and its potential applications

Sarah Jane Hobbs[1], Mark A. Robinson[2] and Hilary M. Clayton[3]

[1] Centre for Applied Sport and Exercise Sciences, University of Central Lancashire, Preston, United Kingdom
[2] Research Institute for Sport and Exercise Sciences, Liverpool John Moores University, Liverpool, United Kingdom
[3] Department of Large Animal Clinical Sciences, Michigan State University, East Lansing, MI, United States of America

## ABSTRACT

**Background.** Ground reaction forces (GRF) measured during equine gait analysis are typically evaluated by analyzing discrete values obtained from continuous force-time data for the vertical, longitudinal and transverse GRF components. This paper describes a simple, temporo-spatial method of displaying and analyzing sagittal plane GRF vectors. In addition, the application of statistical parametric mapping (SPM) is introduced to analyse differences between contra-lateral fore and hindlimb force-time curves throughout the stance phase. The overall aim of the study was to demonstrate alternative methods of evaluating functional (a)symmetry within horses.

**Methods.** GRF and kinematic data were collected from 10 horses trotting over a series of four force plates (120 Hz). The kinematic data were used to determine clean hoof contacts. The stance phase of each hoof was determined using a 50 N threshold. Vertical and longitudinal GRF for each stance phase were plotted both as force-time curves and as force vector diagrams in which vectors originating at the centre of pressure on the force plate were drawn at intervals of 8.3 ms for the duration of stance. Visual evaluation was facilitated by overlay of the vector diagrams for different limbs. Summary vectors representing the magnitude (VecMag) and direction (VecAng) of the mean force over the entire stance phase were superimposed on the force vector diagram. Typical measurements extracted from the force-time curves (peak forces, impulses) were compared with VecMag and VecAng using partial correlation (controlling for speed). Paired samples $t$-tests (left v. right diagonal pair comparison and high v. low vertical force diagonal pair comparison) were performed on discrete and vector variables using traditional methods and Hotelling's $T^2$ tests on normalized stance phase data using SPM.

**Results.** Evidence from traditional statistical tests suggested that VecMag is more influenced by the vertical force and impulse, whereas VecAng is more influenced by the longitudinal force and impulse. When used to evaluate mean data from the group of ten sound horses, SPM did not identify differences between the left and right contralateral limb pairs or between limb pairs classified according to directional asymmetry. When evaluating a single horse, three periods were identified during which differences in the forces between the left and right forelimbs exceeded the critical threshold ($p < .01$).

**Discussion.** Traditional statistical analysis of 2D GRF peak values, summary vector variables and visual evaluation of force vector diagrams gave harmonious results and

Corresponding author
Sarah Jane Hobbs,
sjhobbs1@uclan.ac.uk

both methods identified the same inter-limb asymmetries. As alpha was more tightly controlled using SPM, significance was only found in the individual horse although $T^2$ plots followed the same trends as discrete analysis for the group.

**Conclusions**. The techniques of force vector analysis and SPM hold promise for investigations of sidedness and asymmetry in horses.

## INTRODUCTION

Locomotion results from the application of ground reaction forces (GRF) in accordance with the laws of motion formulated by Sir Isaac Newton. Forces are invisible but they can be measured using multi-dimensional force plates or force shoes. Typically, the 3D force is resolved into components according to the adopted coordinate system that are displayed as force-time graphs. The forces are analyzed by extracting peak values and their times of occurrence, and by calculating the associated impulses. Statistical analyses are then used to detect differences between horses, limbs or experimental conditions.

This approach has facilitated the description of normative GRF values at different gaits (*Clayton, Schamhardt & Hobbs, 2017*; *Khumsap, Clayton & Lanovaz, 2001*; *Merkens et al., 1986*; *Merkens et al., 1993a*; *Merkens et al., 1993b*) and speeds (*Dutto et al., 2004*; *Hobbs, Bertram & Clayton, 2016*). Changes in the typical force patterns and values have been reported in association with lameness (e.g., *Back et al., 2007*; *Ishihara, Bertone & Rajala-Schultz, 2005*; *Khumsap et al., 2003*; *Merkens & Schamhardt, 1988*; *McLaughlin et al., 1996*; *Weishaupt et al., 2004a*; *Weishaupt et al., 2004b*; *Wiestner et al., 2006*; *Weishaupt, 2008*), farriery modifications (e.g., *Riemersma et al., 1996*), local anesthesia (*Bidwell et al., 2004*), asymmetry (*Wiggers et al., 2015*) and training (*Hobbs, Bertram & Clayton, 2016*; *Clayton & Hobbs, 2017*). Left–right asymmetries in peak vertical GRF are the most specific and sensitive GRF variable for detecting lameness (*Ishihara, Bertone & Rajala-Schultz, 2005*) and have been applied in establishing asymmetry thresholds for lameness diagnosis (*Weishaupt, 2008*).

Peak GRF values represent only a small part of the force-time signal and analytic methods based solely on evaluating GRF peaks ignore a large amount of information. A more detailed examination of the GRF curves has been explored using six characteristic time points, five corresponding longitudinal GRF magnitudes and peak vertical GRF (*Dow et al., 1991*; *Williams et al., 1999*). Both studies identified subtle modifications in GRF patterns due to injury or disease and advocated this approach when studying asymmetry. The limitations of using this approach are that a larger number of discrete variables need to be extracted from the force-time curves and as a consequence data analysis and interpretation are also more complicated. In this paper we describe techniques for analyzing continuous force-time data that include all the available information, but with fewer variables and with useful methods to aid visual interpretation. Application of these techniques may assist in

expanding current knowledge of key functional adaptations and in detecting compensatory mechanisms associated with asymmetrical movement patterns.

Visualization of GRF data is based on displaying the force vectors as a series of arrows with the length of each arrow scaled to the magnitude of the force, the orientation indicating the direction in which the force acts, and the origin at the point of force application. In horses, this technique lends itself to displaying the magnitude and direction of the 2D GRFs of one or more limbs in a single diagram, so that differences between horses, limbs and conditions are more easily appreciated. A similar method, known as Pedotti diagrams or Butterfly diagrams, has been used to study 2D GRFs in human locomotion (*Pedotti, 1977*; *Khondadadeh, Whittle & Bremble, 1986*; *Kambhampati, 2007*; *Berki et al., 2015*).

Statistical interpretation of continuous GRF data may also provide an insight into functional differences occurring outside of typical discrete time points (e.g., peak values) within the stance phase (*Pataky, Robinson & Vanrenterghem, 2013*). Here, we present two methods for analyzing the GRF data. Firstly, summary vector variables that encapsulate the effect of the GRF are derived from the sagittal plane vector diagrams for the diagonal limb pairs of trotting horses. These summary vector variables are analysed using traditional statistical methods. Secondly, the continuous vertical and longitudinal GRF vector is explored using statistical parametric mapping (SPM) (*Friston et al., 2007*; *Pataky, Robinson & Vanrenterghem, 2013*), which can objectively identify significantly different regions between two GRF profiles. The analysis in this study is restricted to 2D as the sagittal plane is considered to be the primary plane of motion during trotting (*Merkens & Schamhardt, 1994*), large between and within horse variability have questioned the value of medio-lateral GRF data (*Ishihara, Bertone & Rajala-Schultz, 2005*) and sagittal plane GRF data is still most commonly reported in horses (including *Wiggers et al., 2015*).

The aims of the study are, therefore, firstly to demonstrate the utility of vector diagrams as visual aids that illustrate functional (a) symmetry within horses, secondly to compare the statistical results of traditional discrete GRF variables against (a) summary vector variables and (b) continuous data using SPM analysis in relation to symmetry/asymmetry. Based on the results of *Williams et al. (1999)* it was hypothesized that subtle asymmetries in subjectively sound horses may be detectable using the proposed alternative methods of analysis. As summary vector variables provide metric values that may be used to quantify asymmetry, a third aim was to explore relationships between traditional discrete variables and summary vector variables. It was hypothesized that summary vector variables would be correlated with traditional vector variables, as both sets of variables were obtained from the same GRF data.

## METHODOLOGY

This study was performed with approval from the institutional animal care and use committee under protocol number 02/08-020-00 (Michigan State University, USA).

For the development of the force vector analysis technique, GRF data were collected from 10 horses (mean ± SD, mass = 463 ± 38 kg; height = 1.49 ± 0.07 m). The horses were assessed subjectively as being free from lameness (Grade 0/5 on the American Association

of Equine Pracitioners lameness scale) (*American Association of Equine Practitioners, 2017*) by an experienced veterinarian.

## Data collection

The 10 horses trotted in hand over a series of four synchronized force plates; the first and last plates measured $60 \times 120$ cm (FP60120; Bertec Corp., Columbus, OH, USA) and the middle two measured $60 \times 90$ cm (FP6090; Bertec Corp., Columbus, OH, USA). The rubberized runway was 40 m in length with the force plates embedded in the middle part of the runway and operating at 960 Hz. Retroreflective markers attached to the trunk and mid-lateral wall of each hoof were tracked at 120 Hz using 10 infra-red cameras (Eagle cameras; Motion Analysis Corp., Santa Rosa, CA, USA) and motion analysis software (Cortex 1.1.4.368; Motion Analysis Corp., Santa Rosa, CA, USA). All horses were familiarized with the research environment prior to data collection. Data were collected with the horses trotting at a consistent speed within and between trials until three valid trials had been saved for each horse (speed was derived from the trunk marker). Relative speed was calculated by dividing speed ($ms^{-1}$) by the square root of (standing height (m) multiplied by acceleration due to gravity ($ms^{-2}$)) (*Bertram et al., 2000*). A valid trial consisted of a complete stride that included consecutive clean hits on the force platform from all four limbs. Each stance phase was detected initially using the kinematic data to ensure that only one hoof contacted the force platform, it was completely on the force platform, and was not close to the edge. The stance phases were then identified for each limb using a threshold of 50 N for footstrike and lift off.

Asymmetries in conformation, musculoskeletal adaptations and previous injuries/-pathologies (for example *Pearce, May-Davis & Greaves, 2005*; *Stubbs et al., 2010*; *Wiggers et al., 2015*) amongst other factors may lead to individual gait adaptations that are masked in a group analysis, due to between horse variability. As such, one of the horses (horse 10, mass: 425 kg; height: 1.45 m) was used to explore intra-individual gait asymmetries. In this horse, data collection was continued until eight valid trials had been collected.

## Force Vector Analysis (FVA)

All analyses were conducted using sagittal plane data. To create vector diagrams and summary vector variables GRF data were down-sampled to 120 Hz from the original capture rate of 960 Hz and normalized to horse mass. The GRF vectors of each limb were plotted as a vector diagram in Excel (Microsoft Limited, Reading, Berks, UK). Each vector had its origin at its centre of pressure on the force plate (that is the point of application of the GRF vector), with magnitude and orientation scaled to the sagittal plane GRF, as described by *Kambhampati (2007)*. At a sampling rate of 120 Hz, vectors were created at intervals of 8.3 ms for the duration of the stance phase. The vector diagram was completed by the addition of a scaled vertical reference line passing through the middle of the hoof from heel to toe (Fig. 1).

Quantitative metrics of typical GRF variables were extracted for the group of 10 sound horses based on three trotting trials (one stance phase for each limb per trial, per horse) and for horse 10 individually based on eight trotting trials (one stance phase for each limb

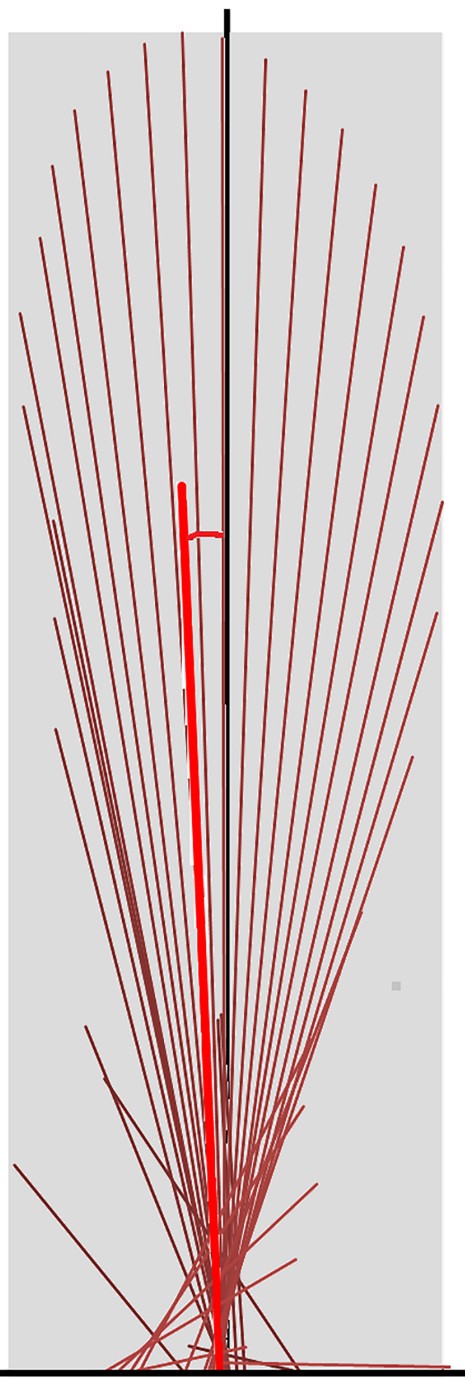

**Figure 1  Force vector diagram for a forelimb of a horse trotting overground.** Sagittal plane force vectors (dark red lines) are constructed at intervals of 8.3 ms. Each vector originates at its centre of pressure, which is identified relative to the hoof location on the force plate. The length is proportional in magnitude to that of the force and it is oriented in the direction of the force. The vector envelope (shaded area) joins the peripheral tips of the force vectors. The black vertical line is drawn through the dorsopalmar midpoint of the hoof. The summary vector is shown as a heavy red line originating at the midpoint of the hoof; its length represents the mean magnitude (VecMag) and its angle to the vertical represents the mean angle (VecAng) of all vectors throughout the stance phase.

for per trial) (*Back et al., 2007*; *Ishihara, Bertone & Rajala-Schultz, 2005*; *Khumsap et al., 2003*; *Merkens et al., 1988*; *McLaughlin et al., 1996*; *Weishaupt et al., 2004a*; *Weishaupt et al., 2004b*; *Wiestner et al., 2006*; *Weishaupt, 2008*). The variables included peak force values, time of force peaks, and  impulses for the vertical, longitudinal braking and longitudinal propulsive GRF in each limb. The shape of the vector diagram was represented by the vector envelope connecting the tips of the vectors that form the vector diagram (Fig. 1). A summary vector representing the magnitude and direction of the mean GRF over the entire stance phase was superimposed on the vector diagram (Fig. 1) and two summary variables were derived; the vector magnitude (VecMag) was calculated by vector summation of the individual vectors divided by the number of samples contributing to the value. The angle of the summary vector (VecAng) was determined from the components of the vector magnitude using trigonometry and expressed relative to the vertical with positive values being directed cranially. Typical GRF variables, and summary vector magnitudes and angles for the fore and hind limbs were tabulated, together with mean speed per step.

Coefficient of variation (COV) of typical GRF variables (peak forces, impulses) and VecMag were calculated and compared using a mean value from the three stance phases for each limb per horse for the group analysis and eight stance phases for each limb for the individual horse analysis. Simple bootstrapping was used on variables that were not normally distributed. Paired samples $t$-tests (left v. right) were performed to compare the values of the discrete and summary vector variables between left and right forelimbs and left and right hind limbs for the group and for Horse 10 individually. Additional analysis of the group data was performed with the diagonal pairs of each horse re-classified by the forelimb mean peak vertical GRF to compare between limbs based on directional asymmetric bias in individual horses (*Starke et al., 2013*). Discrete variables and summary vector variables of the higher forelimb GRF diagonal pair (high) versus the lower (low) forelimb GRF diagonal pair were compared using paired samples $t$-tests (high v. low). Significance was identified at $p < .05$. Relationships between GRF variables and vector variables were explored for the group using partial correlation with relative speed as a covariate.

## Statistical Parametric Mapping (SPM) analysis

For SPM analysis of GRF data, values of the vertical and longitudinal GRF components (mean of three stance phases for each limb and each horse for the group analysis, 8 stance phases for each limb for the individual horse analysis) were normalized to horse mass (N/kg) and stance duration (101 points). For the group analysis, these normalized stance phase values were assembled into 10 * 101 * 2 vector fields (10 horses, 101 data points per stance phase, two dimensions per data point) for each limb. For the individual horse analysis, these normalized stance phase values were assembled into 8 * 101 * 2 vector fields (eight stance phases, 101 data points per stance phase, two dimensions per data point) for each limb. The open-source spm1d package (v. M.0.4.1, *Pataky, 2012*) was used to conduct the SPM analysis in Matlab (R2017a; The Mathworks Inc., Natick, MA, USA). Hotelling's $T^2$ tests, the vector-field equivalent of the paired-samples $t$-test, were used to compare stance phases of the left and right forelimbs and hind limbs (left v. right)

**Table 1  Classification of the group of horses.** Directional asymmetry information from peak forelimb GRF (N/kg) measured for each stance phase for the 10 horses in the group.

| Higher peak forelimb GRF | # horses | # horses with all 3 trials > | # horses with > 4% difference in peak GRF | Range of mean difference in peak forelimb GRF (%) |
|---|---|---|---|---|
| Left | 6 | 4 | 2 | <0.01–5.9 |
| Right | 4 | 3 | 1 | 0.8–3.4 |

using vector GRFs (i.e., vertical and longitudinal continuous data together) for the group of horses and for horse 10 individually (*Pataky, Robinson & Vanrenterghem, 2013*). SPM analysis utilizes Random Field Theory to determine the critical threshold at which only alpha % of equivalently smooth random data would cross. This ensures a tight control of the type I error rate (*Pataky, Vanrenterghem & Robinson, 2016*). Any crossings of the critical threshold therefore by definition have a probability of occurrence less than alpha %. Where significant differences were identified, a post-hoc paired SPM $t$-test was conducted on the two individual GRF components.

Following the initial analysis, the vector fields for each diagonal pair of each horse in the group were then re-classified according to mean peak vertical GRF and compared (high v. low) using SPM.

## Visualization

Qualitative vector diagrams were prepared from an example horse in the group together with typical graphical and force diagrams and stick images. Vector diagrams were initially organized by diagonal limb pairs. Subsequently, diagrams for contralateral limbs were overlaid to facilitate detection of functional differences between left and right limbs.

## RESULTS

For the group of ten horses, mean relative speeds for the two diagonals were: Left Fore Right Hind (LFRH): 0.83 ± 0.10, Right Fore Left Hind (RFLH): 0.83 ± 0.09 (mean ± s.d.). Table 1 reports directional asymmetry bias information for the group of horses. These data were used to re-classify the horses into high and low diagonal pairs. Six horses had higher GRFs in the left forelimb, so in the reclassification according to mean peak vertical GRF in the forelimb, GRF data from four horses moved to the opposite diagonal compared with the left–right classification.

Typical GRF variables and vector summary variables for left and right diagonal pairs and for high and low diagonal pairs for the group are shown in Tables 2 and 3 respectively. COV for the group were lower for vertical GRF components and higher for longitudinal GRF components, except for time to peak propulsive force which had a COV similar to the vertical GRF components. For the summary vectors, variability in VecMag was low (below 10% COV), but the range of VecAng, particularly in the hindlimbs, was relatively large compared to the mean values, which reflects the variability in the longitudinal GRF. Significant differences ($p < .05$) in left compared to right limbs for the group were only found in the forelimbs for peak braking force (left > right), braking impulse (left > right) and vector angle (left more caudally directed than right) (Table 2). In comparison,

**Table 2 Typical force and vector summary variables for the group comparing left diagonal to right diagonal.** Mean, standard deviation (s.d.) and coefficient of variation (COV) ([range] for VecAng) of typical force variables and vector summary variables for left and right fore and hind limbs for the group of 10 horses trotting in hand, overground. Significant differences between left and right limbs are shaded. Simple bootstrapping was applied to variables denoted with an asterisk*.

| Variable | Limb | L/R | Mean (s.d.) | COV (%) [Range] | P value | Bootstrap |
|---|---|---|---|---|---|---|
| Peak vertical GRF (N/kg) | Fore | L | 10.9 (0.73) | 6.7 | .527 | |
| | | R | 10.8 (0.59) | 5.5 | | |
| | Hind | L | 8.6 (0.49) | 5.7 | .476 | |
| | | R | 8.7 (0.74) | 8.5 | | |
| Peak braking GRF (N/kg) | Fore | L | −1.17 (0.19) | 16.2 | .044 | |
| | | R | −1.07 (0.13) | 12.4 | | |
| | Hind | L | −0.70 (0.18) | 25.9 | .251 | |
| | | R | −0.65 (0.07) | 11.0 | | |
| Peak propulsive GRF (N/kg) | Fore | L | 0.87 (0.14) | 15.6 | .083 | |
| | | R | 0.92 (0.12) | 12.9 | | |
| | Hind | L | 1.02 (0.10) | 9.8 | .524 | * |
| | | R | 1.08 (0.10) | 8.8 | | |
| Vertical impulse (Ns/kg) | Fore | L | 1.98 (0.17) | 8.6 | .216 | |
| | | R | 1.95 (0.19) | 9.5 | | |
| | Hind | L | 1.41 (0.12) | 8.7 | .348 | * |
| | | R | 1.43 (0.15) | 10.1 | | |
| Braking impulse (Ns/kg) | Fore | L | −0.12 (0.02) | 12.6 | .020 | |
| | | R | −0.11 (0.01) | 11.5 | | |
| | Hind | L | −0.06 (0.02) | 33.3 | .131 | |
| | | R | −0.05 (0.01) | 14.8 | | |
| Propulsive impulse (Ns/kg) | Fore | L | 0.07 (0.01) | 19.5 | .062 | * |
| | | R | 0.08 (0.01) | 16.5 | | |
| | Hind | L | 0.08 (0.01) | 14.0 | .072 | |
| | | R | 0.09 (0.01) | 15.6 | | |
| Time to peak vertical GRF (% stance) | Fore | L | 45.1 (1.8) | 3.9 | .265 | |
| | | R | 45.6 (2.3) | 5.0 | | |
| | Hind | L | 44.3 (2.3) | 5.1 | .554 | * |
| | | R | 44.7 (1.1) | 2.4 | | |
| Time to peak braking force (% stance) | Fore | L | 28.8 (3.2) | 11.2 | .424 | |
| | | R | 29.4 (4.5) | 15.4 | | |
| | Hind | L | 21.3 (1.5) | 6.8 | .113 | |
| | | R | 20.1 (1.8) | 8.9 | | |
| Time to peak propulsive force (% stance) | Fore | L | 73.7 (1.9) | 3.5 | .401 | |
| | | R | 73.2 (3.1) | 4.6 | | |
| | Hind | L | 63.4 (3.1) | 6.0 | .263 | * |
| | | R | 62.6 (1.9) | 3.7 | | |

**Table 2** (*continued*)

| Variable | Limb | L/R | Mean (s.d.) | COV (%) [Range] | *P* value | Bootstrap |
|---|---|---|---|---|---|---|
| Vector magnitude (N/kg) | Fore | L | 6.47 (0.50) | 7.8 | .345 | |
| | | R | 6.40 (0.42) | 6.6 | | |
| | Hind | L | 4.99 (0.30) | 6.1 | .123 | |
| | | R | 5.09 (0.37) | 7.4 | | |
| Vector angle (degrees) | Fore | L | −1.56 (0.50) | [−0.82, −2.36] | .022 | |
| | | R | −1.09 (0.44) | [−0.13, −1.63] | | |
| | Hind | L | 1.08 (1.04) | [2.39, −0.77] | .103 | * |
| | | R | 1.61 (0.46) | [2.02, 0.80] | | |

significant differences ($p < .05$) for high diagonal compared to low diagonal were only found in the forelimbs for peak vertical GRF (high > low), vertical impulse (high > low) and vector magnitude (high > low), see Table 3.

Figure 1 indicates a typical force vector diagram in which the braking vectors point caudally and delineate the caudal edge of the vector envelope, whereas propulsive vectors point cranially and delineate the cranial edge of the envelope. Figure 2 shows serial stick figures of a trotting horse (Fig. 2A) with the corresponding vertical (Fig. 2B) and longitudinal (Fig. 2C) force-time graphs and force vector diagrams (Fig. 2D) including the summary vectors for the LFRH (left) and RFLH (right) diagonals. The vector diagrams have a preponderance of braking vectors in the forelimbs and propulsive vectors in the hind limbs, and the sign of VecAng is negative in the forelimbs and positive in the hind limbs. The forelimbs typically have a higher peak vertical GRF than the hind limbs resulting in a higher vector envelope and a larger value for VecMag.

Results for SPM of continuous stance phase GRF data for the group of horses are shown in Fig. 3 (left v. right) and Fig. 4 (high v. low) in which the horizontal dashed line indicates the critical threshold above which left and right values are significantly different. There were no significant differences ($p > .05$) in the fore or hind limbs for the analysis combining vertical and longitudinal GRF for either left v. right or high v. low. The *T* value for the forelimbs is a little higher between about 20–70% stance for left v. right, which encompasses the time when both vertical and longitudinal GRF components reach their peak values (Fig. 3). In contrast, the $T^2$ values for the forelimbs for high versus low show a smaller increase over a shorter period of time in the middle of the stance phase, around the time of vertical GRF peak values (Fig. 4).

Results for GRF variables in the individual horse (horse 10) are reported in Table 4. Mean relative speeds for the two diagonals were: LFRH: $0.74 \pm 0.02$, RFLH: $0.74 \pm 0.02$. Within the individual horse, the COV for different variables were similar to those of the group. A number of variables differed significantly between contralateral limbs. In the forelimbs these included peak vertical force which was 2.7% greater in the right forelimb compared to the left, VecMag (right > left) and VecAng (right more caudally directed than left). In the hind limbs, peak vertical GRF, vertical impulse, and peak longitudinal propulsive GRF were higher in the left hind limb.

Results of SPM of stance phase GRF data for horse 10 (Fig. 5) showed significant differences between GRF vectors for the left and right forelimbs ($p < 0.05$) but not for the

**Table 3  Typical force and vector summary variables for the group comparing high GRF diagonal to low GRF diagonal.** Mean, standard deviation (s.d.) and coefficient of variation (COV) ([range] for Ve-cAng) of typical force variables and vector summary variables for high peak forelimb force diagonal and low peak forelimb force diagonal for the group of 10 horses trotting in hand, overground. Significant differences between left and right limbs are shaded. Simple bootstrapping was applied to variables denoted with an asterisk*.

| Variable | Limb | H/L | Mean (s.d.) | COV (%) [Range] | *P* value | Bootstrap |
|---|---|---|---|---|---|---|
| Peak vertical GRF (N/kg) | Fore | H | 11.0 (0.67) | 6.1 | .003 | |
| | | L | 10.8 (0.63) | 5.9 | | |
| | Hind | H | 8.6 (0.65) | 7.6 | .240 | |
| | | L | 8.7 (0.59) | 6.8 | | |
| Peak braking GRF (N/kg) | Fore | H | −1.16 (0.19) | 16.5 | .061 | |
| | | L | −1.08 (0.13) | 12.0 | | |
| | Hind | H | −0.66 (0.09) | 13.6 | .657 | |
| | | L | −0.68 (0.17) | 25.7 | | * |
| Peak propulsive GRF (N/kg) | Fore | H | 0.89 (0.13) | 14.4 | .772 | |
| | | L | 0.90 (0.13) | 14.6 | | |
| | Hind | H | 1.03 (0.09) | 8.9 | .293 | |
| | | L | 1.07 (0.11) | 10.1 | | |
| Vertical impulse (Ns/kg) | Fore | H | 1.99 (0.18) | 9.1 | .005 | |
| | | L | 1.93 (0.17) | 8.8 | | |
| | Hind | H | 1.42 (0.14) | 9.8 | .615 | |
| | | L | 1.43 (0.13) | 9.0 | | |
| Braking impulse (Ns/kg) | Fore | H | −0.12 (0.02) | 15.2 | .074 | |
| | | L | −0.11 (0.01) | 8.3 | | |
| | Hind | H | −0.05 (0.01) | 19.8 | .789 | |
| | | L | −0.05 (0.02) | 34.2 | | |
| Propulsive impulse (Ns/kg) | Fore | H | 0.07 (0.01) | 18.7 | .662 | |
| | | L | 0.08 (0.01) | 18.1 | | |
| | Hind | H | 0.09 (0.01) | 13.6 | .494 | |
| | | L | 0.09 (0.02) | 17.1 | | |
| Time to peak vertical GRF (% stance) | Fore | H | 45.3 (1.87) | 4.1 | .765 | |
| | | L | 45.4 (2.20) | 4.9 | | |
| | Hind | H | 44.5 (2.29) | 5.1 | .926 | * |
| | | L | 44.5 (1.07) | 2.4 | | * |
| Time to peak braking force (% stance) | Fore | H | 28.8 (3.70) | 12.8 | .514 | |
| | | L | 29.3 (4.17) | 14.2 | | |
| | Hind | H | 20.4 (2.20) | 10.8 | .408 | |
| | | L | 21.0 (1.02) | 4.8 | | |
| Time to peak propulsive force (% stance) | Fore | H | 73.7 (2.31) | 3.1 | .238 | |
| | | L | 73.1 (2.83) | 3.9 | | |
| | Hind | H | 63.1 (2.63) | 4.2 | .841 | |
| | | L | 62.9 (2.51) | 4.0 | | |

**Table 3** (*continued*)

| Variable | Limb | H/L | Mean (s.d.) | COV (%) [Range] | *P* value | Bootstrap |
|---|---|---|---|---|---|---|
| Vector magnitude (N/kg) | Fore | H | 6.52 (0.47) | 7.2 | .006 | |
| | | L | 6.35 (0.45) | 7.1 | | |
| | Hind | H | 5.03(0.38) | 7.5 | .795 | |
| | | L | 5.05 (0.31) | 6.2 | | |
| Vector angle (degrees) | Fore | H | −1.45 (0.67) | [−0.13, −2.36] | .273 | |
| | | L | −1.20 (0.28) | [−0.82, −1.63] | | |
| | Hind | H | 1.28 (0.52) | [2.02, 0.50] | .718 | |
| | | L | 1.41 (1.08) | [2.39, −0.77] | | |

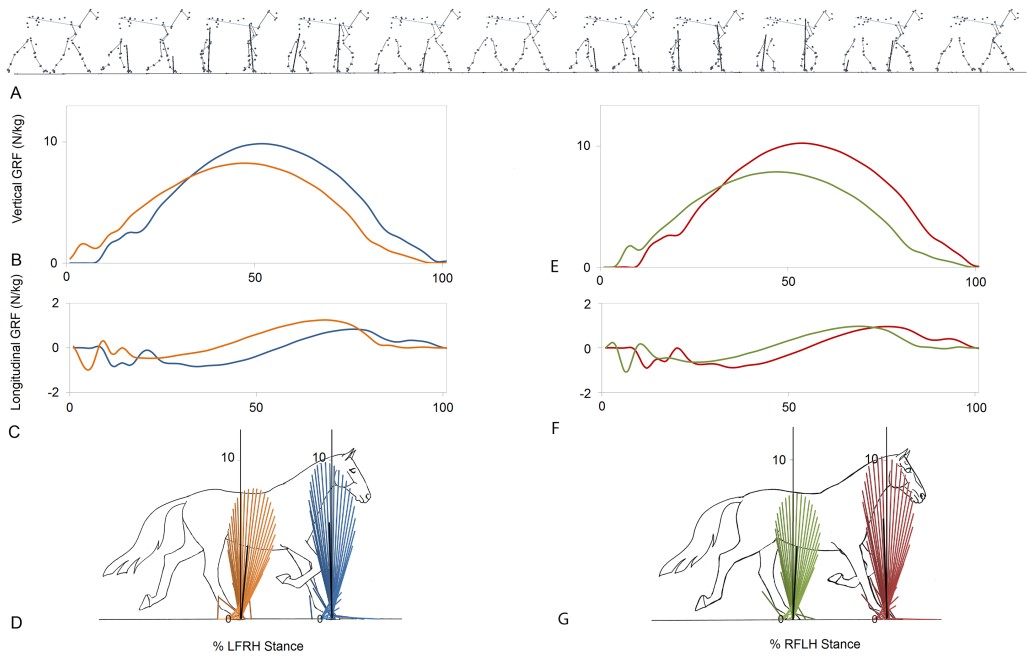

**Figure 2** **Stick figures, time-normalized vertical and longitudinal ground reaction forces and vector diagrams for the two diagonals of one consecutive stride from one horse trotting overground.** (A) Sequential stick figures of the horse at 10% intervals of the stride. (B, E) Vertical ground reaction forces (N/kg) for the fore and hind limbs of each diagonal pair. (C, F) Longitudinal ground reaction forces (N/kg) for the fore and hind limbs of each diagonal pair. (D, G) Vector diagrams constructed from the force data in (B, E) and (C, F) showing summary vectors for each limb (summary vectors shown in black). Blue, left forelimb; red, right forelimb; orange, right hind limb; green, left hind limb. RH, right hind; LF, left fore; LH, left hind; RF, right fore; LFRH, left fore and right hind diagonal; RFLH, right fore and left hind diagonal.

hind limbs. In the forelimbs three clusters of data points exceeded the critical threshold at 14.8–16.2%, 43.0–45.9% and 83.7–85.1% stance (all $p < .001$). Post-hoc analysis of the individual GRF components revealed that the first two peaks in the 2D vector result are primarily influenced by the vertical component whereas the third peak is likely a combination, but more by the longitudinal component.

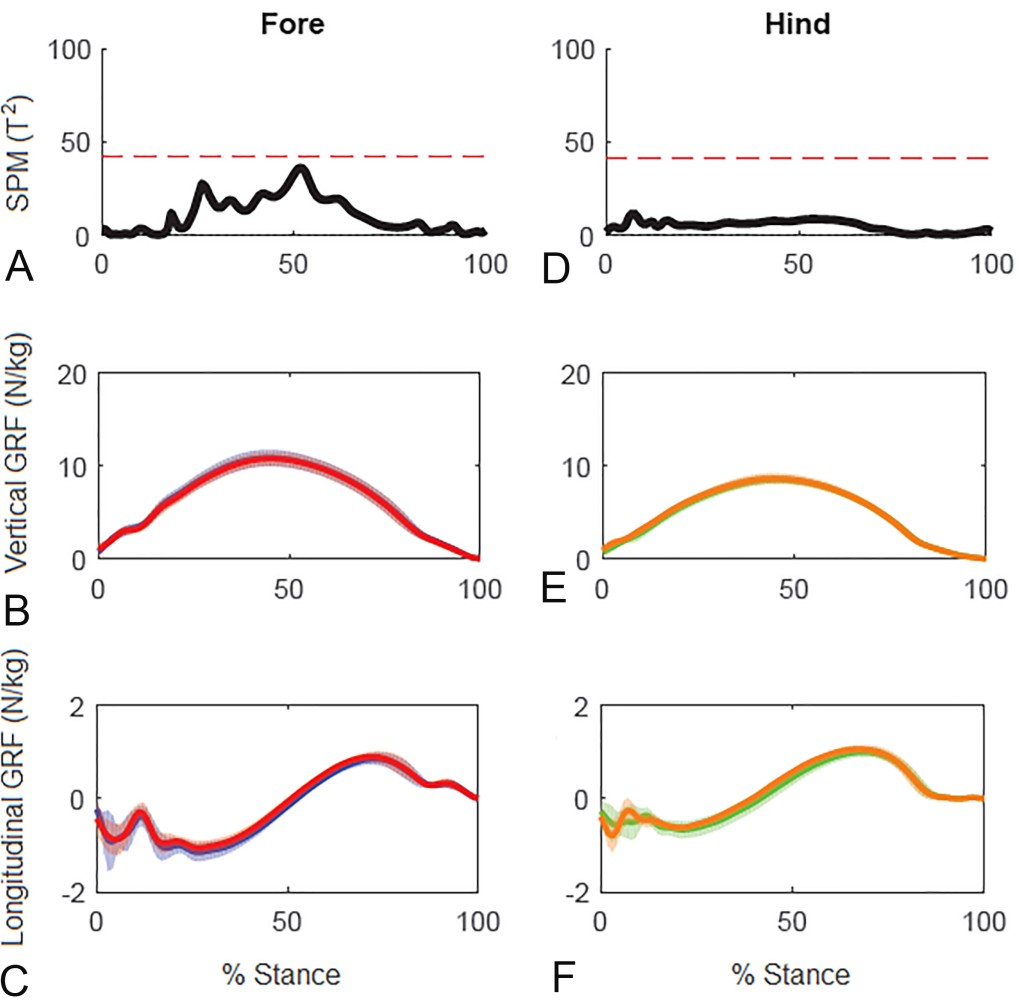

**Figure 3 Results of statistical parametric mapping of ground reaction forces at trot overground for a group of 10 horses (mean of 3 stance phases per horse, per limb).** (A, D) The Hotelling's $T^2$ test results (vertical and longitudinal ground reaction forces combined), the $T^2$ statistic (black line) did not exceed the critical threshold (red dashed line) therefore there were no statistical differences. (B, E) mean (solid line) and standard deviation (shaded area) vertical and (C, F) mean (solid line) and standard deviation (shaded area) longitudinal ground reaction forces for the group (N/kg). Blue, left forelimb; red, right forelimb; orange, right hind limb; green, left hind limb.

Superimposition of the vector diagrams for the contralateral limbs (Fig. 6) show differences in function of the left and right limbs during one stride in horse 10. The right forelimb envelope is taller (corresponding with the second period of SPM significance), but not noticeably wider than that of the left forelimb. The shape of the envelope and density of the vectors indicate that the right forelimb produces more braking GRF in early stance whereas the left forelimb produces more propulsion in late stance, which coincide with the first and third clusters of data points that reached significance in the SPM plot (Fig. 5). The left hind limb has a taller and wider envelope than the right hind limb in this trial, but

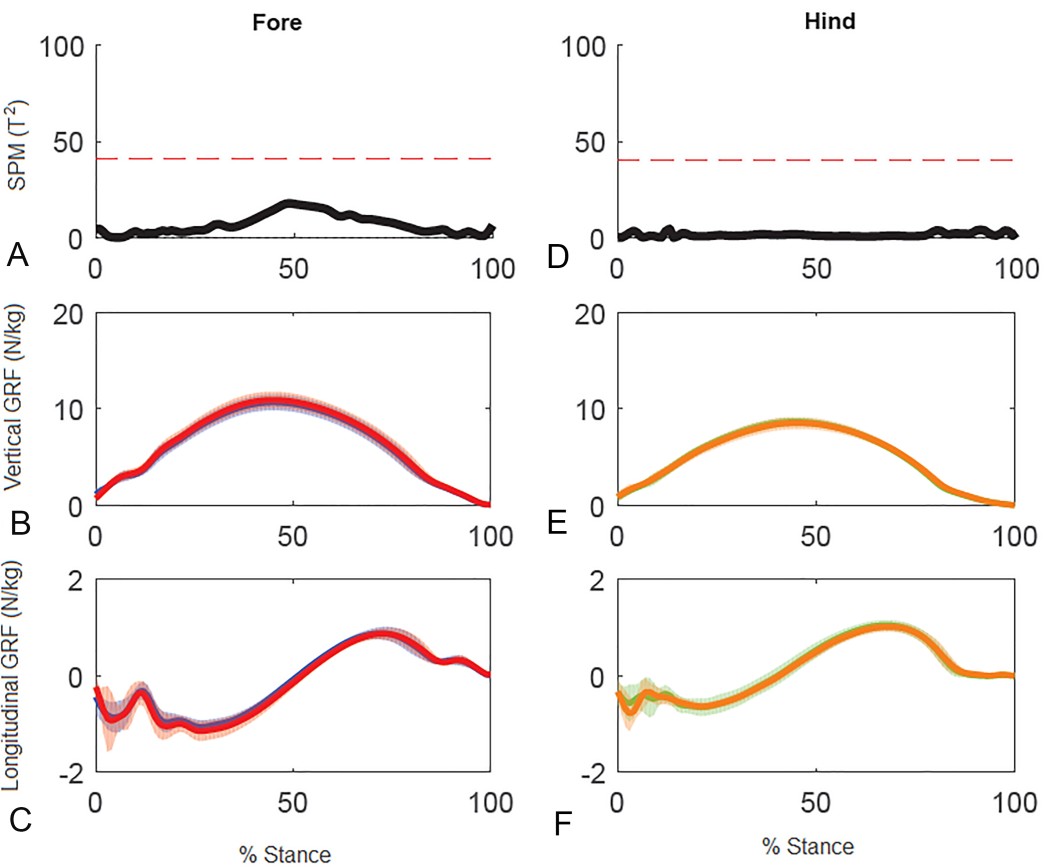

**Figure 4 Results of statistical parametric mapping of ground reaction forces of trot overground for a group of 10 horses when categorized by higher and lower forelimb peak vertical GRF (mean of 3 stance phases per horse, per limb).** (A, D) The Hotelling's $T^2$ test SPM results (vertical and longitudinal ground reaction forces combined), the $T^2$ statistic did not exceed the critical threshold (red dashed line) therefore there were no statistical differences. (B, E) mean (solid line) and standard deviation (shaded area) vertical and (C, F) mean (solid line) and standard deviation (shaded area) longitudinal ground reaction forces for the group (N/kg). Red, forelimb with higher peak vertical GRF; Blue, forelimb with lower peak vertical GRF; orange, diagonal hind limb of the forelimb with higher peak vertical GRF; green, diagonal hind limb of the forelimb with lower peak vertical GRF.

as vector and SPM analysis found no significant differences between hindlimbs, this only illustrates between limb variability that may occur.

The statistical relationships between traditional GRF variables and vector summary variables (Table 5) indicate that VecMag was most strongly related ($R > .9$) with peak vertical forces (forelimbs $p < .001$, hindlimbs $p < .001$). For the forelimbs, VecMag was also strongly ($R > .7$) related with vertical impulse ($p < .001$), peak braking force ($p < .001$), and braking impulse ($p < .001$). The strongest relationships for VecAng were with peak braking force ($R > .6$; forelimbs $p = .004$, hindlimbs $p < .001$), peak propulsive force in the forelimbs ($R > .6$; forelimbs $p = .005$), braking impulse ($R > .6$; forelimbs $p = .006$, hindlimbs $p < .001$) and propulsive impulse ($R > .6$; forelimbs $p = .002$, hindlimbs $p = .001$).

**Table 4  Typical force and vector summary variables for the individual horse.** Mean, standard deviation (s.d.) and coefficient of variation (COV) ([range] for VecAng) of typical force variables and vector summary variables for left and right fore and hind limbs for horse 10 trotting in hand, overground. Significant differences between left and right limbs are shaded. Simple bootstrapping was applied to variables denoted with an asterisk*.

| Variable | Limb | L/R | Mean (s.d.) | COV (%) [Range] | P value | Bootstrap |
|---|---|---|---|---|---|---|
| Peak vertical GRF (N/kg) | Fore | L | 10.36 (0.18) | 1.71 | <.001 | |
| | | R | 10.64 (0.22) | 2.09 | | |
| | Hind | L | 9.21 (0.28) | 3.07 | .012 | |
| | | R | 9.03 (0.20) | 2.22 | | |
| Peak braking GRF (N/kg) | Fore | L | −1.02 (0.09) | 9.12 | .052 | |
| | | R | −1.13 (0.08) | 7.16 | | |
| | Hind | L | −0.73 (0.06) | 8.00 | .374 | |
| | | R | −0.76 (0.07) | 8.72 | | |
| Peak propulsive GRF (N/kg) | Fore | L | 1.04 (0.07) | 6.40 | .051 | |
| | | R | 0.96 (0.09) | 9.44 | | |
| | Hind | L | 1.12 (0.10) | 9.16 | .008 | * |
| | | R | 1.00 (0.08) | 7.47 | | |
| Vertical impulse (Ns/kg) | Fore | L | 2.19 (0.03) | 1.26 | .018 | |
| | | R | 2.22 (0.03) | 1.32 | | |
| | Hind | L | 1.62 (0.05) | 1.26 | .614 | |
| | | R | 1.62 (0.03) | 1.96 | | |
| Braking impulse (Ns/kg) | Fore | L | −0.13 (0.01) | 6.44 | .024 | |
| | | R | −0.14 (0.01) | 8.43 | | |
| | Hind | L | −0.06 (0.01) | 17.36 | .093 | |
| | | R | −0.07 (0.01) | 14.87 | | |
| Propulsive impulse (Ns/kg) | Fore | L | 0.10 (0.01) | 9.53 | .016 | |
| | | R | 0.09 (0.01) | 10.73 | | |
| | Hind | L | 0.11 (0.01) | 11.44 | .019 | |
| | | R | 0.09 (0.01) | 11.04 | | |
| Time to peak vertical GRF (% stance) | Fore | L | 44.09 (1.25) | 2.84 | .520 | * |
| | | R | 44.51 (1.66) | 3.74 | | |
| | Hind | L | 44.24 (1.57) | 3.54 | .164 | * |
| | | R | 45.88 (1.92) | 4.18 | | |
| Time to peak braking force (% stance) | Fore | L | 25.32 (2.36) | 9.31 | .436 | |
| | | R | 25.98 (0.74) | 2.87 | | |
| | Hind | L | 19.75 (1.93) | 9.79 | .824 | |
| | | R | 19.98 (2.13) | 10.64 | | |
| Time to peak propulsive force (% stance) | Fore | L | 72.24 (1.59) | 2.19 | .090 | |
| | | R | 71.19 (1.27) | 1.78 | | |
| | Hind | L | 66.03 (2.00) | 3.03 | .024 | |
| | | R | 69.40 (1.47) | 2.12 | | |
| Vector magnitude (N/kg) | Fore | L | 6.10 (0.10) | 1.67 | .001 | * |
| | | R | 6.35 (0.16) | 2.50 | | |
| | Hind | L | 5.15 (0.27) | 5.32 | .269 | |
| | | R | 5.08 (0.16) | 3.15 | | |

**Table 4** (*continued*)

| Variable | Limb | L/R | Mean (s.d.) | COV (%) [Range] | *P* value | Bootstrap |
|---|---|---|---|---|---|---|
| Vector angle (degrees) | Fore | L | −0.67 (0.48) | [−0.14, −1.35] | .016 | |
| | | R | −1.47 (0.48) | [−0.89, −2.30] | | |
| | Hind | L | 1.51 (0.74) | [2.95, 0.47] | .057 | |
| | | R | 0.67 (0.75) | [1.94, −0.35] | | * |

## DISCUSSION

This study illustrates the use of a potentially visually appealing and intuitive technique that (1) condenses the large amount of information and numerous variables generated by traditional GRF analysis into a single diagram with two summary variables, VecMag and VecAng, and (2) describes a method for analyzing multi-dimensional continuous data. Comparisons of between limb function by superimposition of their vector diagrams may be easier to visualize than more traditional GRF curves and differences between the diagrams can be tested statistically using summary vector variables. In addition, a better understanding of functional adaptations throughout the stance phase may be achieved by SPM of continuous data. Subtle asymmetries in subjectively sound horses were detected using the vector analysis and SPM, but only significantly for the individual horse using SPM. The hypothesis can therefore be accepted, but variability and directional bias in a group will affect the threshold limits and therefore level of significance using SPM. Summary vector variables strongly correlated with traditional variables, so this hypothesis was accepted.

Visual evaluation of force vector diagrams has previously been described and applied in human biomechanical analysis under the name Pedotti diagram, which refers to the researcher who initially described the method (*Pedotti, 1977*; *Kambhampati, 2007*; *Marasović, Cecić & Zanchi, 2009*) or butterfly diagram, which refers to the fact that a walking person generates a vector envelope that is shaped like a butterfly wing (*Khondadadeh, Whittle & Bremble, 1986*; *Berki et al., 2015*). We have used the more generic term force vector diagram, which is applicable to all gaits and all species. The potential advantages of force vector analysis over traditional GRF analysis are that the diagrams may be easier to read and interpret by clinicians and researchers who do not have detailed knowledge of biomechanics. They may also have practical applications in evaluating gait asymmetries, and for monitoring changes over time in response to training or rehabilitation.

The force vectors originate at the centre of pressure (COP) beneath the hoof and a greater vector density indicates slower progression of the COP under the hoof (*Nauwelaerts, Hobbs & Back, 2017*), whereas more widely separated vectors occur when the COP is moving more rapidly. Since the end point of each force vector is determined by the magnitude and direction of the applied force, the vector envelope has a typical and different shape for the forelimb and hind limb in accordance with their different functional responsibilities. In trot the envelopes have the shape of a vertically-oriented ellipse. The envelope is taller and more caudally oriented in the forelimbs in accordance with the higher vertical and braking forces compared with the hind limbs which have a shorter and more cranially oriented envelope due to the higher propulsive forces. In the forelimbs but not the hind limbs,

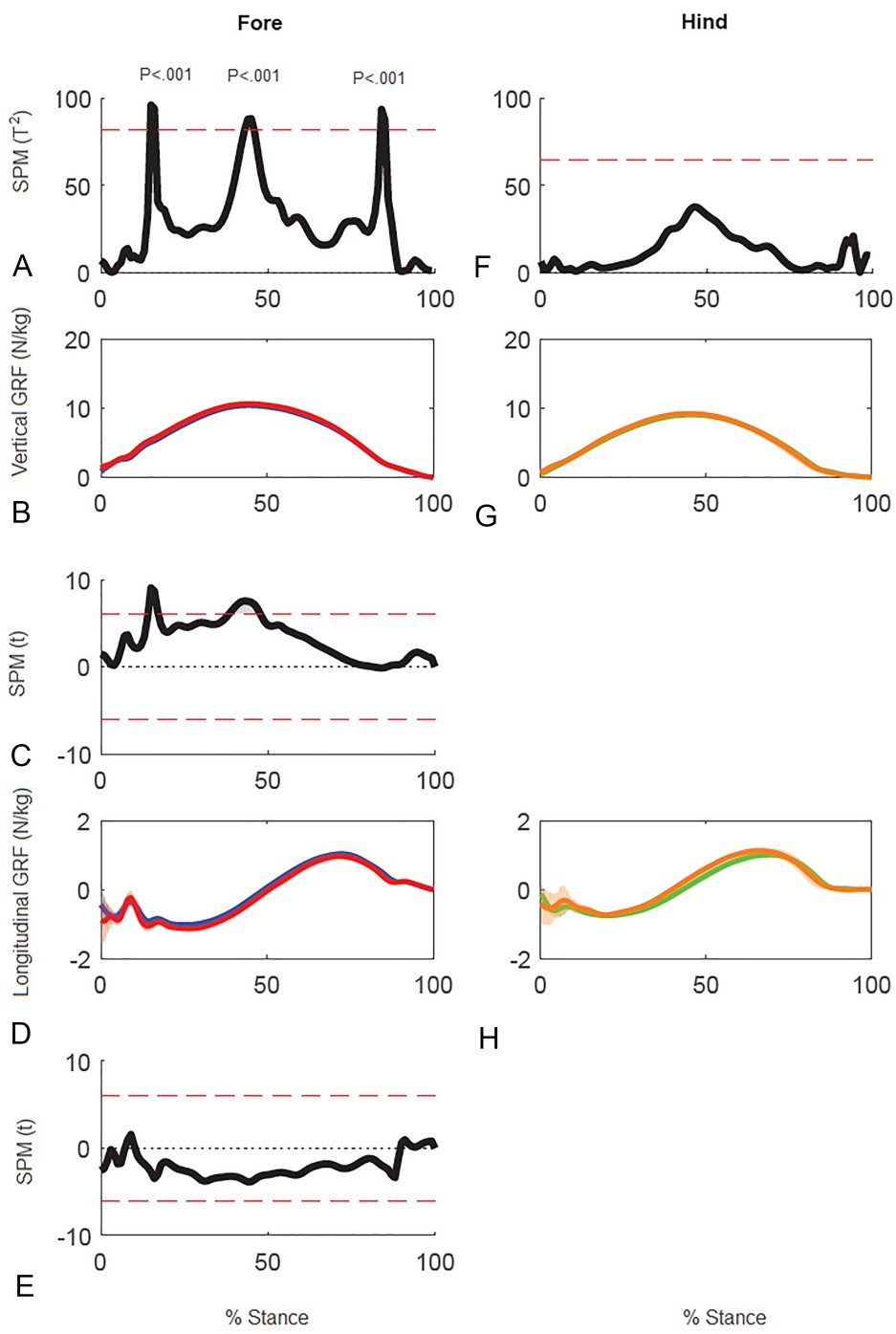

**Figure 5** **Results of statistical parametric mapping of ground reaction forces of trot overground for one horse (8 stance phases per limb).** (A, F) The Hotelling's $T^2$ test SPM results (vertical and longitudinal ground reaction forces combined), the $T^2$ statistic crossed the critical threshold in three regions indicating that there was a significant differences in the forelimbs, (B, G) mean (solid line) and standard deviation (shaded area) vertical ground reaction forces (N/kg), (C) post-hoc paired-samples SPM(t) results for the vertical force component, (D, H) mean (solid line) and standard deviation (shaded area) longitudinal ground reaction forces (N/kg), and (E) post-hoc paired-samples SPM(t) results for the longitudinal force component for horse 10 Blue: left forelimb; red, right forelimb; orange, right hind limb; green, left hind limb.

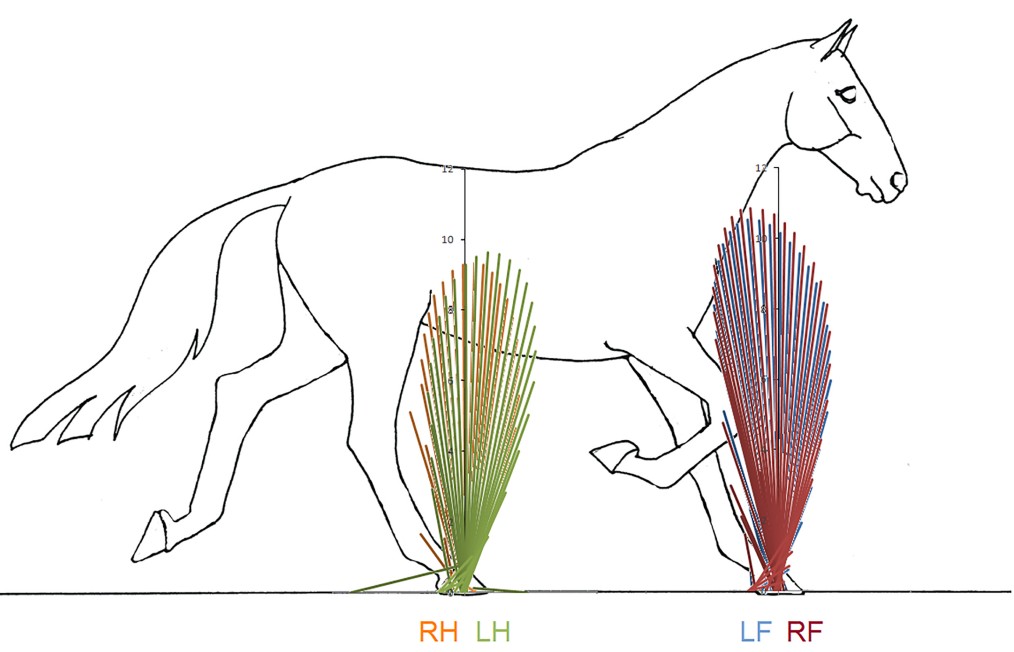

**Figure 6  Force vector diagrams overlaid for contralateral limbs of horse 10 trotting overground.** RH, right hind; LF, left fore; LH, left hind; RF, right fore.

the height of the envelope is expected to increase with trotting speed as a consequence of the increase in forelimb vertical force (*Dutto et al., 2004*). Therefore, it is important to consider the effects of speed when comparing force vector diagrams and to investigate to what extent steady state locomotion was performed over the analysed stride cycle.

Evaluating force vector analysis and SPM against traditional methods of analysis, data for the left–right and high–low comparisons for the group provided an insight into similarities and differences in findings that may be expected. Greater braking GRF and impulse and a more caudally aligned VecAng in the left forelimb when comparing left to right is illustrated in $T^2$ plots coming closer to the significance threshold in SPM for the forelimb. In contrast, when comparing the higher GRF diagonal pair to the lower, only the vertical GRF, impulses and VecMag in the forelimb were significant, which is indicative of the re-classification of the diagonal pairs. For SPM of high–low, $T^2$ plots were further from the significance threshold and only closer around the time of peak GRF in the forelimbs. Two possible approaches could explain the differences in significance between methods. Either traditional and vector analysis yielded false positives, unlike SPM analysis which uses random field theory to tightly control alpha (*Pataky, Vanrenterghem & Robinson, 2016*). Alternatively, functional asymmetry in braking GRFs between left and right forelimbs is evident and may relate to sidedness, but this is masked when grouping the diagonals by vertical GRFs. In humans, dominant limbs (dominant being the right limb if right handed) are able to make well directed, smooth and energy efficient movements, whereas non-dominant limb movements tend to be less efficient (*Sainsburg, 2014*). In horses, evidence of sidedness is emerging, but habitual postural and locomotor preferences

Peer J

**Table 5  Relationships between typical force variables and vector summary variables for the group.** Partial correlation matrix between vector metrics and typical force variables for the group of 10 horses (controlling for relative speed). Bootstrapped variables are identified with the following symbol §. Significant relationships are highlighted; $p < .05^*$, $p < .01^{**}$ and shaded.

| | Peak vertical GRF F | Peak vertical GRF H | Peak braking GRF F§ | Peak braking GRF H | Peak propulsive GRF F | Peak propulsive GRF H§ | Vertical impulse F | Vertical impulse H§ | Braking impulse F | Braking impulse H | Propulsive impulse F | Propulsive impulse H | Vec Mag F | Vec Mag H | Vec Ang F | Vec Ang H |
|---|---|---|---|---|---|---|---|---|---|---|---|---|---|---|---|---|
| Peak vertical GRF F | 1.000 | | | | | | | | | | | | | | | |
| Peak vertical GRF H | .637** | 1.000 | | | | | | | | | | | | | | |
| Peak braking GRF F§ | −.691** | −.352 | 1.000 | | | | | | | | | | | | | |
| Peak braking GRF H | .135 | −.063 | −.161 | 1.000 | | | | | | | | | | | | |
| Peak propulsive GRF F | .364 | .319 | −.128 | −.418 | 1.000 | | | | | | | | | | | |
| Peak propulsive GRF H§ | .061 | .199 | .326 | .037 | .416 | 1.000 | | | | | | | | | | |
| Vertical impulse F | .741** | .535* | −.637** | −.121 | .699** | .150 | 1.000 | | | | | | | | | |
| Vertical impulse H§ | .326 | .465* | −.130 | −.338 | .640** | .317 | .663** | 1.000 | | | | | | | | |
| Braking impulse F | −.670** | −.408 | .924** | −.037 | −.207 | .241 | −.737** | −.324 | 1.000 | | | | | | | |
| Braking impulse H | .120 | .093 | −.111 | .929** | .442 | .012 | −.226 | −.445 | .030 | 1.000 | | | | | | |
| Propulsive impulse F | .218 | .204 | −.091 | −.466* | .959** | .347 | .688** | .630** | −.180 | −.530* | 1.000 | | | | | |
| Propulsive impulse H | .386 | .368 | −.183 | .223 | .467 | .319 | .505* | .733** | −.290 | .144 | .404 | 1.000 | | | | |
| VecMag F | .939** | .764** | −.738** | .114 | .387 | .041 | .792** | .370 | −.739** | .120 | .265 | .372 | 1.000 | | | |
| VecMag H | .758** | .905** | −.577** | .205 | .221 | .104 | .674** | .421 | −.613** | .253 | .146 | .436 | .868** | 1.000 | | |
| VecAng F | −.317 | −.142 | .632** | −.406 | .615** | .468* | .001 | .276 | .610** | −.412 | .665** | .111 | −.338 | −.348 | 1.000 | |
| VecAng H | .311 | .244 | −.183 | .812** | −.022 | .202 | .133 | .110 | −.138 | .800** | −.128 | .701** | .299 | .406 | −.210 | 1.000 |

are commonly linked to dorsal hoof wall angle differences (*Van Heel et al., 2006*). Greater braking forces have been reported in hooves with a flatter dorsal hoof wall angle, but usually in combination with an increase in peak vertical force (*Wiggers et al., 2015*). As such, these results do not consistently suggest that the group had a flatter hoof angle in the left forelimb, but this may not preclude evidence of functional sidedness. As asymmetric bias is of interest in horses, particularly when considering sub-clinical conditions, the methods of analysis used in future studies should be an important consideration until we have a clearer understanding of common functional adaptations.

The values for VecMag in both forelimbs and hind limbs were predominantly correlated with forelimb peak forces and impulses for the vertical and braking GRF components. The force plate and the measured GRF variables have been described as the gold standard in lameness diagnosis (*Keegan et al., 2011*), with peak vertical GRF having the highest sensitivity and specificity for classifying horses as lame versus sound (*Ishihara, Bertone & Rajala-Schultz, 2005*). These authors suggest that a 7% decrease in peak vertical GRF is equivalent to an increase of 0.5/5.0 in the subjectively-assessed lameness grade as described by the *American Association of Equine Practitioners (2017)*. Since VecMag represents the magnitude of limb loading over the entire stance phase, it is anticipated that between-limb differences in VecMag combined with the vector diagrams could be valuable for detecting and interpreting functional asymmetry. *Keegan (2007)* highlighted the fact that some lameness conditions, notably mild superficial digital flexor tendinopathy and navicular disease, are associated with a change in shape of the vertical GRF curve rather than simply a reduction in peak vertical GRF in the lame limb (*Williams et al., 1999*). In these cases, the shape change involved a decrease in vertical GRF in the early (superficial digital flexor tendinopathy) or later (navicular disease) part of stance, with the timing being related to the function of the affected structures. Changes of this nature should be visible on vector diagrams and detectable using SPM, and may be interpreted in light of knowledge about which structures are active at different times during the stance phase. Thus, the shape of the vector diagram and the timing of asymmetries detected by SPM might contain potentially useful information to aid in localizing some physiological adaptations that have functional consequences.

The SPM method was initially used for analysis of brain images (*Friston et al., 2007*). However, the ability to detect changes in time-series data lends itself to the analysis of biomechanical data, as functional adaptations may not be fully captured when the analysis is confined to discrete variables representing single events (*Pataky, Robinson & Vanrenterghem, 2013*). SPM analysis might prove to be useful for detecting asymmetries in the individual horse due to sidedness. This was illustrated in horse 10 which was assessed clinically as being sound. Even though peak vertical force differed significantly between the forelimbs (due to a directional bias), the magnitude of the difference (2.7%) was less than the threshold value used to detect subclinical lameness on the treadmill (*Wiestner et al., 2006*). Therefore, it is not surprising that the horse was subjectively assessed as sound. In this horse SPM revealed three periods of asymmetry in the forelimbs. If these periods of asymmetry can be related to specific events in the stride cycle, then it may facilitate interpretation of their functional significance. When biomechanical data are averaged over

a group of horses, much of the individual variability tends to be lost. In order to further the understanding of locomotor asymmetries, for example sidedness due to cerebral laterality, it might prove useful to evaluate horses on an individual basis and to explore asymmetry throughout the stride rather than at discrete time points. The results present a preliminary indication that SPM may prove useful for this purpose.

In horse 10 SPM identified three time periods when the forces differed significantly between left and right forelimbs coinciding with secondary impact (*Gustås, Johnston & Drevemo, 2006*), peak vertical force (*Hobbs & Clayton, 2013*), and heel lift (*Merkens & Schamhardt, 1994*). Therefore, it seems plausible that there are functional inter-limb differences at these times. In general the results of traditional force plate analysis, force vector analysis and SPM analysis are in agreement. For example, the mid-stance difference between left and right forelimbs identified by SPM is consistent with the difference in forelimb peak vertical force identified by traditional statistical analysis and with the difference in VecMag identified by the force vector diagrams and analysis (Fig. 6).

In the forelimb and hind limb, VecAng is highly correlated with the peak longitudinal forces and longitudinal impulses indicating that it represents the longitudinal GRF effects. High variability in longitudinal GRFs has previously been reported in non-lame horses and horses with low grade (1/5) lameness, which led to the conclusion that longitudinal GRFs were not useful for quantifying lameness grade (*Ishihara, Bertone & Rajala-Schultz, 2005*). The between- and within-horse variability reported in that study concur with our findings and may reflect the need for the horse to adjust its balance from stride to stride (*Clayton & Hobbs, 2017*). The between-limb differences are particularly striking when the vector diagrams for left and right limbs are overlaid to show the relative lengths and angles of the vectors (Fig. 6).

At steady-state gait one might expect the summed vertical impulses for the diagonal pairs to be equal and the longitudinal impulses summed over both diagonal pairs to be zero (*Hobbs & Clayton, 2013*; *Lee, Bertram & Todhunter, 1999*). The functional role of the forelimbs is to provide a mechanism to alter the centre of mass (COM) trajectory from forwards and downwards at diagonal contact to forwards and upwards at diagonal lift off (*Bertram & Gutmann, 2009*). In horses, the large forelimb vertical impulses elevate the COM (*Weishaupt et al., 2009*; *Clayton, Schamhardt & Hobbs, 2017*; *Clayton & Hobbs, 2017*) and increase nose-up pitching moments (*Hobbs, Bertram & Clayton, 2016*). Asymmetries between left and right limbs and diagonals may or may not be repeated consistently from step to step, as they reflect the need for subtle adjustments of balance within a single stride (*Hobbs, Bertram & Clayton, 2016*). This is particularly evident in the hindlimbs, as no significant differences were found in the group analyses. Therefore, asymmetries in force vector diagrams that are present inconsistently should be considered with caution with respect to lameness.

Limitations exist in using force vector analysis and SPM over more traditional methods. Force vector analysis, in particular VecMag is sensitive to the method used to define the stance phase, as the vector is divided by the number of samples (see Supplementary Information). It is therefore important to report the threshold used to define the stance phase and it would be preferable to adopt a standard threshold in the future. For SPM, due

to the tight control of alpha, reaching significance may not be as important as understanding the clinical implications of functional adaptations seen throughout the entire stance phase. This should be taken into consideration when using SPM to study asymmetry within groups of horses.

## CONCLUSION

This study illustrates the application of force vector analysis as a visual tool that can be used to evaluate and compare GRFs between limbs and horses. The present study is a 2D analysis in the sagittal plane but can be applied to other 2D planes and extended to 3D data. Two summary vector variables that represent the overall effect of the applied force during the entire stance phase are presented; VecMag correlates with the vertical force and impulse and VecAng correlates with the longitudinal force and impulse. Vector diagrams are used to display and compare sagittal plane GRFs throughout stance. Statistical analysis of the force vectors using Statistical Parametric Mapping shows promise as a tool to detect subtle inter-limb asymmetries at defined time periods that correspond with specific kinematic events.

### Funding
The authors received no funding for this work.

### Competing Interests
The authors declare there are no competing interests.

### Author Contributions
- Sarah Jane Hobbs conceived and designed the experiments, analyzed the data, contributed reagents/materials/analysis tools, prepared figures and/or tables, authored or reviewed drafts of the paper.
- Mark A. Robinson analyzed the data, contributed reagents/materials/analysis tools, prepared figures and/or tables, authored or reviewed drafts of the paper.
- Hilary M. Clayton conceived and designed the experiments, performed the experiments, analyzed the data, contributed reagents/materials/analysis tools, prepared figures and/or tables, authored or reviewed drafts of the paper.

### Animal Ethics
The following information was supplied relating to ethical approvals (i.e., approving body and any reference numbers):

This study was performed with approval from the institutional animal care and use committee under protocol number 02/08-020-00 (Michigan State University, USA).

### Data Availability
The raw data are provided in the Supplemental Files.

## Supplemental Information

Supplemental information for this article can be found online at http://dx.doi.org/10.7717/peerj.4399#supplemental-information.

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
