# Peer review of "A simple method of equine limb force vector analysis and its potential applications"

_PeerJ, doi:10.7717/peerj.4399_

## Round 0.1 · original submission · Major Revisions

I thank the authors for this interesting MS and the two reviewers for public reviews that are constructive and reasonable. The reviewers essentially agree that there is merit to the study but they provide plenty of useful points to address in revising the MS, especially reviewer 1. Please ensure to address all points individually in a Response document. There are several requests for more statistical tests (or tweaks thereof) and I think it will be necessary to do some of these. The MS will need re-review. Thank you and we look forward to seeing the resubmission.

·

Basic reporting

Basic reporting: This manuscript aims to investigate the use of time force vector diagrams captured over the stance phases of horses for illustrating functional asymmetry and to compare traditional ground reaction force values (peak force, impulse etc) to continuous data in this context. The manuscript is generally very well written and presented. Figures/Tables are relevant and well described.

Experimental design

Experimental design: The research aims are defined. There are however no research hypothesis presented. Including specific objectives/hypotheses may help the reader in terms of for example easier understanding of which parts of the investigation have been performed to study left-right differences and which parts are investigating ‘force mechanisms’, i.e. the general relationship between peak force, impulse etc and the newly proposed continuous parameters (VecMag, VecAng).
No concerns about Ethics: non-invasive procedures during standard in-hand exercise which is normal routine for many horses. Technical standard high: combined multi-force plate and 3D motion capture data.
Methods: Generally appropriate level of detail. Could be more comprehensive in some places, particularly when describing the statistical method for comparing continuous data. Sample sizes should be presented more accessibly: it is not immediately clear what the relationship is between the number of ‘trials’ and the number of ‘stance phases’. Please report sample sizes (number of stance phases, not number of trials) when reporting results of statistical tests.

Validity of the findings

Validity: The main difficulties encountered when reading this manuscript are listed below (in order of perceived importance) followed by some more detailed comments referencing specific sections in the manuscript:
1. In the context of functional asymmetry analysis (lameness, as listed as one of the motivators for this study by the authors), population level comparison of left-right asymmetry is of comparatively little use when the direction of asymmetry is not dealt with appropriately. Let me give a more detailed explanation: assume five of the ten horses show smaller force left fore than right fore and the other five show smaller force right fore and the magnitude of the measured force asymmetries (independent of what force parameters is chosen) are similar. A paired sample assessment of LEFT versus RIGHT GRF values will in all likelihood show no significant difference since negative and positive values (i.e. differences between left and right values) will cancel out in the analysis. It is hence questionable what a population level analysis applied to the traditional left right values and also applied to the suggested continuous time data contributes to a comparison between the two approaches. The authors may need to think about ways of dealing with this effect e.g. using some sort of ‘normalization’ of the effect of different ‘directions’ of asymmetries. I understand that here the authors are dealing with perceivedly sound horses (some more direct evidence about this assumption, e.g. direct reporting of peak force and impulse differences (in percent?) for each horse may be helpful in this context).
The above challenge with left-right differences does not affect the conducted correlation analysis, which (if I understand correctly) has been applied to the individual limb values rather than to left-right differences. If the authors however aim at correlating asymmetry values between the traditional and the continuous approach, then further thought needs to go into the normalization of the direction of asymmetry.
2. The discussion puts (in comparison to the population level data) quite a lot of emphasis on the ‘pilot data’ of one particular horse for which eight trials (rather than 3 trials, how may stance phases?) has been collected and for which the proposed temporal analysis is identifying some significant left-right differences at specific times over the stance phase. This is being used to discuss in detail the advantages of using this analysis in the context of lameness despite the horse not having been identified as lame. This speculative part may benefit from some condensing and from a more balanced discussion of this topic.
It may also be interesting to analyse in a bit more detail the influence of the algorithm used to identify beginning and end of stance. In particular over the first and last approximately 10% of the stance phase, force values are changing rapidly so I slight difference in how the left and right limb contact the ground may cause a temporal shift that may lead to the detection of significant differences in this transient phase of the force signal. A sensitivity analysis investigating the influence on changes in foot on and foot off timings may provide information about the robustness of the statistical approach towards these temporal effects and then justify a slightly more detailed discussion of this part of the study. A recent study (by one of the co-authors of the current manuscript) indicates that a difference of 26ms captures 95% of the differences between kinetic and kinematic timings for foot on and foot off events. This would indicate a possible shift of about 3 samples (for the sub-sampled 120Hz data). It would appear very interesting to provide some information about the robustness of the statistical method applied to the continuous data against time shifts of this magnitude.
3. The correlation analysis between the traditional and the continuous parameters is very nice (please add p-values to the provided correlation values in the main text). The authors may want to consider discussing the difficulty of comparing a novel method (providing complementary information) to a more limited but established method. A correlation analysis of course shows how well the two are agreeing (so in one way it is good if they do). However if there is perfect agreement and the aim is to create some sort of improvement over the established method: why is the new method needed when it agrees perfectly so does not provide new information? It may be worth discussing how is it possible to show that the new method provides something that the established one doesn’t? Just a thought.

Additional comments

Line 60-62 and following paragraph (line 63-67): thank you for providing relevant references indicating the use of peak vertical ground reaction force for lameness detection. There is also some evidence that peak vertical ground reaction force may be less useful than more intricate analysis based on principal component analysis of different force parameters, which may indeed help making an argument for additional ways of analysing force data. Williams, Silverman, Wilson, Goodship, 1999, AJVR 60, 549-555 and also the seminal work that shows that tendon injuries in Thoroughbreds may be detected with force plate analysis (Dow, Leendertz, Silver, Goodship, 1991, EVJ, 23, 266-272). One of the studies is indeed referenced in the discussion.

Line 81-84: I am wondering whether it may be worth introducing (or contrasting to) the method of using ‘functional limits of agreement’ which has been used for statistical comparison of IMU data over entire stride cycles here? While not applied to force data over a stance phase, this also appears to be a method that can be used for kinematic data over entire stride cycles. (Olsen, Pfau, Ritz, 2013, J Biomech, 46, 2320-2325). My understanding is, that the method proposed here can deal with multi-dimensional time continuous data while the previously published method cannot?

Line 85-89: Did you have any research hypotheses (and objectives) linked to the aims and objectives? I have struggled at that point in the manuscript to get my head round which analysis actually looks at asymmetries (compares left to right) and which parts is hunting a ‘mechanism’, ie correlation between established and new values (is that latter done on left-right comparisons?

Line 94-97: given that the authors mention the use of peak force asymmetry thresholds for lameness detection it would appear useful to learn more about the peak force asymmetry values for the horses. Testing for statistical differences between left and right may not be a useful undertaking (table 1) here, since some horse may show reduction in left and some horses reduction in right peak force and hence no statistically significant difference may be detectable with this approach. Please consider alternative methods for providing information about force asymmetries between limbs. This relates to main issue 1 outlined above.

Line 106-107: Please specify what you mean by valid trials in this context. How many foot strikes were available from each trial? One per force plate per limb? What were the conditions that had to be met for a ‘valid’ trial. Wherever possible, it would be nice to mention the sample size (in figures/tables and where results are presented). In the current version, it was not clear to me what the sample size was (N=3 trials? N=8 trials?). It would appear more intuitive to me to expess sample size as the number of foot strikes recorded.

Line 108-110: This may need some more ‘motivation’ here both in terms of why intra-horse gait asymmetry was further investigated and why in one horse only. I would consider this a worthwhile more speculative part of the investigation (in particular if robustness against time shifts in identifying stance phases from kinematic data is included). A little more background may help the reader.

Line 115-117: This may be my limited understanding but can you explain why you use the term centre of pressure and not the term point of zero moment which I am more familiar with in the realm of force plate analysis as opposed to pressure mat analysis where the term centre of pressure appears more logical (to me).

Line 118: express in ms (despite ‘s’ being the SI unit) to make more efficient use of decimal places?

Line 121-124/line 136-138: please clarify how many foot strikes (stance phases) were obtained per trial.

Line 139-141: this relates to the point raised earlier about the effect of ‘left-sided’ and ‘right-sided’ asymmetries in force cancelling out. It is also not quite clear to me what these t-tests are aiming for: which of the study aims is the addressing? Consider being more explicit about this.

Line 141-143: Please clarify what variables (e.g. for each limb? Left-right comparisons?) this part of the analysis was performed for.

Line 179-184 (and Figure 3): I am no entirely sure that I completely understand what has been tested for here: is this (in analogy to the t-tests testing for left-right differences) to establish whether there is a difference in the 2D continuous stance phase GRF? If this is the case, could the authors please add some more detail (to the introduction) explaining the rationale of testing for left-right differences on population level, taking into account that some horses may show left-sided bias and some show right-sided bias so the significance of the ability (or failure) to detect any left-right difference is not quite clear? One point in the introduction mentioned as an application of force plate analysis (with the traditionally used parameters) is for example to identify lameness. Comparing ‘population level’ left-right differences between the traditional (peak force, impulse etc) and the time continuous approach appears of limited information value in the context of lameness when not dealing with side/direction of asymmetry in some way or other.

Line 193-196: this is very exciting in the context of lameness analysis (following the topic above and chosen as one of the topics in the introduction) but some additional information may be useful here. Can you please clarify how many stance phases are included in this analysis (is the number of trials equal to the number of stance phases, presumably not since more than one force plate has been used?). This may also be useful to understand why the value of the ‘critical threshold’ is different for the forelimb and hind limb data. Please consider giving a little more information about this threshold in the materials and methods section.

Line 197-201: Please add significance values (p-values) for the reported correlations in the text.

Line 224-227: Please indicate clearly that these are the authors’ opinions and do not relate to findings that you have tested for in this study (study evaluated for example correlation between the two techniques but not whether the results were ‘easy to read’ or ‘intuitive’). Add ‘potential for’ to the last sentence mentioning lameness, rehabilitation.

Line 238-239: … and to investigate to what extent steady state locomotion was performed over the analysed stride cycle?

Line 264-277: This appears to be a somewhat speculative section based on one horse assessed as ‘clinically sound’. It seems plausible (to me) that this technique might indeed have some uses. However, it may be prudent to be a little more ‘balanced’ in these speculations and for example replace the phrase ‘it will be necessary to … explore asymmetry throughout the stride …’ by a phrase such as ‘it might prove useful to …’ and ‘the results presented here suggest … is well suited’ by ‘the results presented here present some indication that SPM may be suitable for this purpose’. Consider robustness/sensitivity analysis against time shifts.

Line 278-297: Even more so than the paragraph before, this section is based on the results of one horse categorized as ‘clinically sound’ be veterinary expert analysis as well as by a suggested force threshold. Please consider considerably shortening or removing this section. It appears a little ‘dangerous’ to discuss this (in the current length and depth) based on the data of one horse without clinical indication without debating the underlying issue of decision making in this context and the difficulty of improving decision making based on the limitations of current best practice. This topic appears immanently relevant in the context of this study which is motivated by saying ‘current force plate analysis is limited’ and we investigate a method that may have potential in this context. The current analysis shows that the new method shows some correlations with the established method hence in effect ‘shows the same’. It appears important (to me) to debate how progress can be made.

Line 319-320: Please restrict the conclusion section as much as possible to reporting the direct consequences of this study. For example: how does this study show that the method can be used to compare GRFs between ‘conditions’. What conditions are you referring to?

Line 322: I would suggest being more specific here and indicating that ‘VecMag correlates to vertical force (and impulse)’ and ‘VecAng to longitudinal forces and impulses’

Line 323: The study has not tested for ‘intuitiveness or simplicity’

Table 1+2 and materials and methods (Coefficient of variation): a coefficient of variation for an angle may be somewhat of an odd thing to do? If you choose a different ‘zero point’ or for example an opposite direction of your angle definition, an angle of +1 degrees varying by +/-0.1 degrees will have a 10% variation while choosing to express this angle as let’s say -359 degrees will in a value of 0.03% variation. Are there better ways of doing this? Is a CoV useful for angles or is an angle not already ‘normalized’.

Figure 1: what are the grey (non-straight) lines?

Figure 2: add shaded area indicating stride-to-stride variation to panels B and C? what is the sample size (N=? stance phases)

Figure 3: sample size (N=? stance phases, not trials if they are not the same). This may need reconsidering since this relates to main issue 1 outlined above with regards to directionality of left-right differences at population level.

Thank you very much for providing raw data in the form of Excel spreadsheets that are reasonably well labelled with respect to their content. It is still not easy to comprehend all of this but I guess it never is when dealing with a data set collected and processed by somebody else. Well done.

·

Basic reporting

The manuscript compares different techniques to analyze ground reaction forces during stance phase of horses when trotting.
Although this matter has been addressed in human biomechanics, it is interesting in the context of animal biomechanics.

Experimental design

Experimental design is fine. Indeed, it is very good.

Validity of the findings

As I have mentioned above, although this matter has been addressed in human biomechanics, it is interesting in the context of animal biomechanics.

Additional comments

The manuscript compares different techniques to analyze ground reaction forces during stance phase of horses when trotting.
Although this matter has been addressed in human biomechanics, it is interesting in the context of animal biomechanics.
The manuscript is well written and explained.
I have only some minor comments and suggestions.

Abstract
Line 17 and Line 19
Please, indicate here what was compared (contralateral fore and hind limbs).

Line 31
“In addition SPM was performed to seek differences between combined vertical and longitudinal GRF throughout the stance phase.”
This sentence leads the reader to conclude that vertical GRF was compared to longitudinal GRF, although this is not the case.
I suggest rewriting this sentence according to Line 34: “differences between the left and right contralateral limb pairs”.

Line 48
“vertical, longitudinal and transverse components…”
As this paragraph has general comments about ground reaction forces, I think these terms are not appropriated since they apply only for quadrupeds (at least the term “longitudinal”. For humans, longitudinal is, indeed, the vertical component). Please, use more general terms, such as “Typically, the 3D force is resolved into components according to the adopted coordinate system…”, or indicate that you are referring to quadrupeds.

Lines 85-89
Please, justify why you have done the analysis only in sagittal plane.

Line 135
Please, explain how mean speed was calculated.
Line 143
Please, define “relative speed” and explain how it was calculated.

Lines 160-161
Please, define the acronyms LFRH and RFLH. I suppose they are left forelimb/right hindlimb and right forelimb/left hindlimb, respectively, but I think, for clarity, it is necessary to define them.

Line 176
Please, consider improving the quality of the coordinate axis in Figure 2. They are too light. It is difficult to see VecAng is negative in the forelimbs and positive in the hindlimbs, especially in Fig. 2-D. Some guys, like me, have to use awfull glasses.

Line 182
I think it is not exactly the “SPM value” that is shown in Fig. 2 and 3. It is the t2 value. Consider rewriting this sentence.

Line 193
Figure 4 – Consider to include post-hoc tests here, for vertical and longitudinal GRF components. I think they would help to understand what was discussed in Discussion section.

Line 204
The parentheses were not closed.

Lines 207 and 208
I suggest to rewrite this sentence with something as “which coincide with the first and third clusters of data points that reached significance in SPM plot (Figure 3).”

Lines 208-209
This sentence is somewhat ambiguous. Consider rewriting this sentence.

Discussion section
As I have previously mentioned, I think a plot showing post-hoc tests for longitudinal and vertical GRF components in Figure 4 would help to understand/visualize several points in the Discussion section.

Line 310
Please, define the acronym “COM”.

Line 322
“…VecMag represents the vertical force and VecAng represents the longitudinal force.”
I suggest rewriting this sentence as “…VecMag is more related to (or more influenced by) the vertical force and VecAng is more related (or more influenced by) the longitudinal force.”

Conclusion
I suggest including a comment that vector diagrams and SPM analysis can be extended to other planes, and SPM, indeed, can be performed in 3D considering simultaneously the three GRF components (and, indeed, vector diagrams can also be performed in 3D: see Fig. 1 in Pataky et al. Vector field statistical analysis of kinematic and force trajectories. Journal of Biomechanics, 46 (2013), 2394-2401).

Table 3 has several acronyms that were not defined. Besides, I suggest presenting the variables in the same order they were presented in Tables 1 and 2.

Figure Captions
Figure 1: include a description for the gray lines.

Figure 2: include a description for the acronyms RH, LF, LH, RF, LFRH, and RFLH

Figure 5: include a description for the acronyms RH, LH, LF, and RF.

---

## Round 0.2 · accepted · Accept

Congratulations-- both reviewers are more than satisfied with the revised MS and I agree! One has the good point that dimensionless speeds should not have units. Please finalize the MS and it will be published shortly!

·

Basic reporting

I have no further comments on this manuscript. The authors have dealt more than adequately with my initial comments.
The figures and tables have been improved considerably.

Experimental design

No further comments. Experimental design is appropriate.

Validity of the findings

No further comments. Inclusion of higher versus lower comparison has alleviated my initial concerns with regard to asymmetry and lameness.

Additional comments

Thanks for taking into consideration the original comments. Much appreciated.

One small point: I believe the reported relative speeds (results section) should be dimensionless and not given in m/s.

·

Basic reporting

No additional comments.

Experimental design

No additional comments.

Validity of the findings

No additional comments.

Additional comments

The manuscript have improved substantially. All my suggestions were included and all my comments were properly reponded. I have no more comments. Congratulations!